# Influence of a Landslide on a Tunnel in Loess-Bedrock Ground

Zhijie Sun [1,2,3], Xuanyu Yang [2,3], Shuai Lu [1], Yang Chen [1] and Pengfei Li [1,*]

1    The Key Laboratory of Urban Security and Disaster Engineering, Ministry of Education, Beijing University of Technology, Beijing 100124, China; 18835130490@163.com (Z.S.); shuai_lu@foxmail.com (S.L.); cybfcc@163.com (Y.C.)
2    Shanxi Transportation Technology Research & Development Co., Ltd., Taiyuan 030032, China; yxy_sxti@163.com
3    Shanxi Laboratory of Intelligent Transportation, Taiyuan 030032, China
*    Correspondence: lpf@bjut.edu.cn

**Abstract:** By combining model testing and numerical simulation, this paper focuses on the influence of landslides on tunnels in loess-bedrock strata by using the perfect landslide–tunnel system (LTS). A mechanical test and simulation (MTS) system was used to provide thrust for loading and unloading the trailing edge of the slope. A Particle Image Velocimetry (PIV) and 32 cluster strain gauges were adopted to monitor the deformation of the tunnel structure and landslide soil, and the sliding surface, respectively. By means of a numerical simulation, the deformation characteristics of a tunnel crossing loess-bedrock strata are comprehensively described. The influence of a cyclic load on the mechanical behavior and displacement of the tunnel and sliding surface is discussed in detail. The experimental results show that the thrust required for the first landslide is the largest, during multiple loading and unloading. With the increase in loading and unloading time, the sliding thrust gradually decreases and eventually remains stable. The landslide presents a progressive failure mode. There is a stress concentration in the upper part of the tunnel, which causes the secondary sliding phenomenon. The deformation of the sliding surface mainly occurs in the upper soil of the tunnel. The deformation direction of the tunnel is consistent with the sliding direction, and the deformation of the sliding surface mainly occurs in the soil above the tunnel. When disturbed by an external force, the tunnel deforms downward, and, when unloaded, the tunnel has a small rebound deformation. However, with the increase in loading–unloading times, the rebound deformation of the tunnel gradually decreases, and the permanent deformation gradually accumulates until the tunnel fails. The research results can provide reference for the construction and protection of tunnel engineering in loess regions, and have reference value for the control of tunnels crossing landslides.

**Keywords:** tunnel; loess-bedrock landslide; model test; deformation characteristics; stick-slip

## 1. Introduction

Loess-bedrock landslides are one of the main types of landslides. The huge impulse generated during sliding and the large volume of such landslides are very likely to cause rock instability, which is highly hazardous [1]. The deformation of a loess-bedrock landslide is affected by many factors, and the deformation characteristics are different under different conditions. The bedrock layer in contact with the loess forms over a short term and is mainly composed of sedimentary rocks [2], which contain sandstone and mudstone. These rocks are susceptible to water ingress. Once the rock is softened, the anti-slide force of the interface between the loess and bedrock decreases and a slide occurs [3–6]. In addition, water that permeates into the loess will make the soil self-weight increase, meaning the slide force increases. In addition, in the back edge of a slope, a series of tension cracks provide a path for water infiltration. Water in these back edge cracks will introduce pore water pressure and seepage pressure, which are key factors that induce slides. Furthermore, vibrational effects due to seismic or engineering activities can also induce some landslides [7,8].



Landslides lead to stress conditions for tunnels, and the stress state and deformation characteristics of a tunnel–landslide system are different for each landslide. There are several kinds of factors that influence tunnel structures: the spatial relationship, the slide area behavior, the soil behavior, the tunnel structure parameters, etc. [9–11]. The model test and numerical simulation are regarded as the most effective methods to investigate the deformation characteristics of a Landslide–Tunnel System (LTS). Although numerical simulation offers an effective and low-cost method, it encounters difficulties in determining the material parameters. The parameters of the soil obtained from model tests or field tests or variation of the envelope parameters or parameter sensitivity analysis are usually used to determine the parameters for numerical simulations to study the impact of landslides on a tunnel [12–14]. The model test can provide the material parameters and the real deformation situation, so the two methods are usually combined to achieve this aim. Many model tests are employed in the context of LTS, including model box, vibrostand, and centrifuge systems. The stress and displacement of the slope during the tunnel excavation process have been studied, considering the creep of slip-face soil under water-softened conditions [15]. In addition, seismic activity is one of the key factors affecting the stability of the second lining in shallow-buried tunnels [16,17]. The loess behavior and tunnel burial depth both have an impact on tunnel stability [18]. The spatial relationship between a tunnel and landslide has also been studied using a model box, where the tunnel axial was the same as the slope dip-direction [19–21]. What is more, the collapse of the foundation under a tunnel affects the tunnel stability [22]. There have also been a series of model experiments examining tunnels and slopes [23–25].

In most studies, the deformation and mechanical behavior are the primary focuses. In geological history, loess deposited on bedrock may have moved many times. Loess-bedrock landslides exhibit a special form of movement, which is called stick-slip. With the tunnel excavation, the loess is disturbed and moves again. This movement form is a potential threat to tunnel stability. Therefore, there are two key problems in LTS research: the stick-slip characteristics of the slope and mechanical behavior of the tunnel. In order to solve these problems, this paper developed a slip surface deformation monitoring method through model testing of LTS and monitored the lateral deformation of the slope surface through particle image velocimetry (PIV). The research reveals the stick-slip mechanism of the landslide and the mechanical behavior of the tunnel, which enriches present understandings of LTS and is of great significance for improving tunnel stability.

## 2. Methodology

The soil for the model test was taken from the landslide body at the engineering site. The model tests need to satisfy geometric similarity, kinematic similarity, dynamic similarity, and physical similarity in terms of the materials. The geometric similarity ratio was obtained by the size ratio of prototype to model, $C_l = l_p/l_m$. There are 10 related physical quantities in this model test, as shown in Equation (1), including geometric size ($l$), Poisson's ratio ($\mu$), friction angle ($\varphi$), density ($\rho$), permeability coefficient ($k$), moisture content ($\omega$), cohesion ($c$), elastic module ($E$), gravity acceleration ($g$), and time ($t$). According to the second theorem of similarity, the similarity criteria are derived by dimensional analysis, which is based on the geometric size, density, and gravitational acceleration.

$$\pi_1 = w; \pi_2 = \varphi; \pi_3 = \frac{c}{l\rho g}; \pi_4 = \frac{k}{l^{1/2}g^{1/2}};$$
$$\pi_5 = \mu; \pi_6 = \frac{E}{l\rho g}; \pi_7 = \frac{t}{l^{1/2}g^{-1/2}} \tag{1}$$

Since the test soil was taken from the landslide site, the similarity ratio of density, elastic modulus, cohesion, and permeability coefficient is 1. The similarity ratio for the dimensionless quantities is 1. In addition, the similarity ratio of geometric size is 40 and the time is 6.3. The properties of the materials used in the laboratory tests are shown in Table 1.

**Table 1.** Basic physical properties of materials.

| Natural Density $p$/(g/cm³) | Natural Moisture Content $w$/(%) | Specific Gravity $\gamma$ | Liquid Limit $W_L$/(%) | Plastic Limit $W_P$/(%) | Grain Size Grading/(%) | | | Cohesion $c$/(kPa) | Friction Angle $\varphi$/(°) |
|---|---|---|---|---|---|---|---|---|---|
| | | | | | >2 mm | 2~0.075 mm | <0.075 mm | | |
| 1.67 | 11.2 | 2.65 | 31 | 18 | 0.3 | 32.1 | 67.6 | 17 | 28 |

### 2.1. Model Preparation

The model was used to simulate a tunnel crossing a loess-bedrock landslide, where cement was used to simulate bedrock with a certain stiffness and a smooth surface. Model similarity ratio tests were performed to ensure material and dimensional similarity.

The sample of loess was collected from the upper part of the collapsed area. The material was predominantly silt (<0.05 mm), with good structure, compactness, and uprightness. The final ratio of materials was determined by the direct shear test and bending test. The particle size of the sand was 1 mm, and the content was 25%, while the loess content was 60%, and the talc content was 15%. After each soil sample was prepared, it was placed in a sealed box for 48 h to ensure uniform soil moisture content. A large number of white particles were incorporated within the soil as markers. The slope gradient was 30° with an 80 cm height, 200 cm length, and 100 cm width. The diameter of the tunnel was 20 cm, and the lining thickness was 1 cm. The tunnel and lining were prepared with gypsum and geogrid. There were 6 parts, and each part was 20 cm long. Epoxy resin was used to connect the tunnel. A tunnel entrance wall was prepared with a geometric ratio 1:60. The model size is shown in Figure 1.

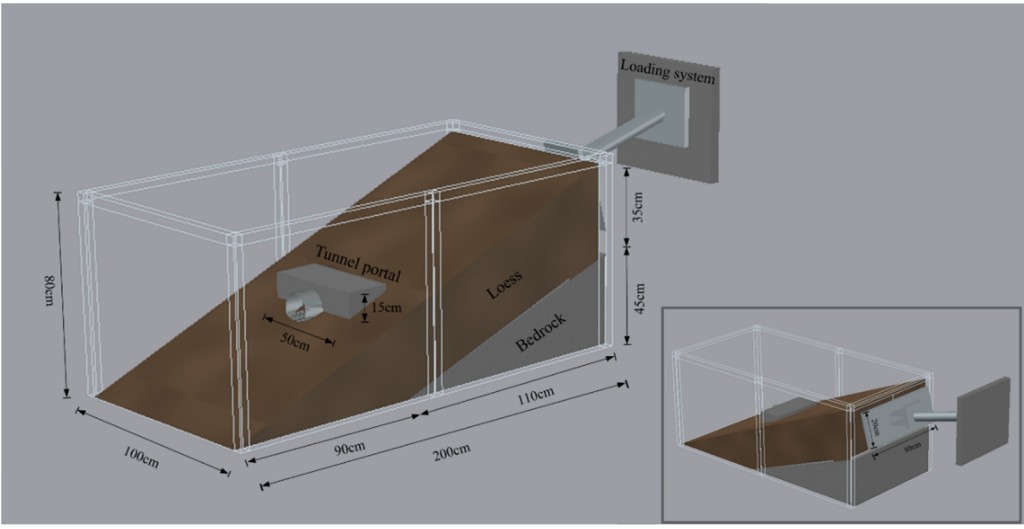

**Figure 1.** Schematic of the test model.

### 2.2. Test Methods

The slope box model was fastened to the shear wall, and the MTS loading system on the shear wall was used to provide the force on the trailing edge of the slope (Figure 2). The MTS is controlled by oil pressure, which can provide 10 tons of force. The loading system accuracy is 0.1 mm and 0.1 kN. The slope sliding was simulated by force control, and the loading mode comprised repeated loading and unloading. This experiment adopted the method of repeated loading to simulate stick-slip. Firstly, the load rate was 0.2 kN/min for 30 min. Once the maximum force reached 60 kN, the force was immediately unloaded. The unloading state was maintained for 20–30 min until the displacement was stable. After this, the next loading stage was conducted. The critical slip point was determined by force—Displacement curve. At the same time, the tunnel deformation was measured by comparing the laser cross positions at different times.

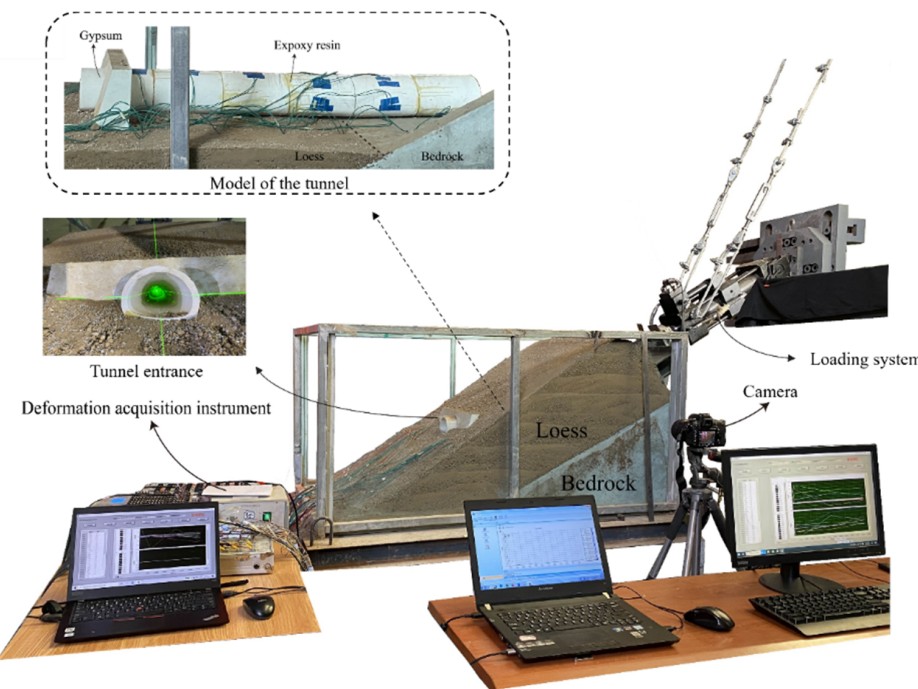

**Figure 2.** LTS model and loading method.

The slide face deformation was monitored by 32 strain gauges. There were 7 rows (1, 2, 3, 4, 5, 6, 7) and 5 columns (A, B, C, D, E) to set strain gauges on the interface (Figure 3a). There were only two strain gauges in the 6th row to avoid the tunnel. The strain gauges were inserted into the bedrock to a depth of 10 mm, and the displacement of each strain gauge was measured by micrometer caliper. The soil pressure was measured using an earth pressure cell (Figure 3b), and the loess was divided into 3 layers: slope bottom (1-1, 1-2, 1-3), tunnel bottom (2-1, 2-2, 2-3, 2-4, 2-5, 2-6, 2-7, 2-8, 2-9), and tunnel roof (3-1, 3-2, 3-3).

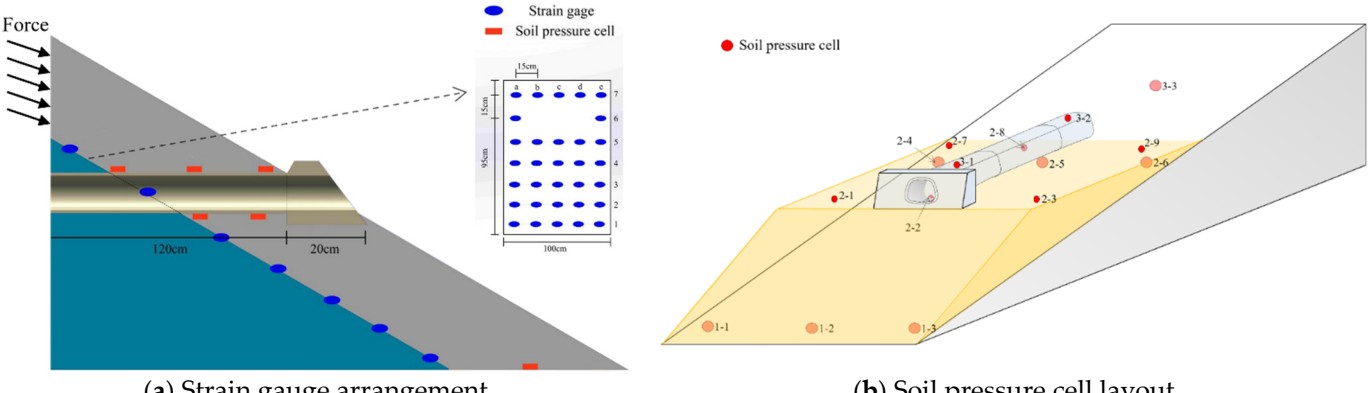

(**a**) Strain gauge arrangement        (**b**) Soil pressure cell layout

**Figure 3.** Diagram of soil pressure cell layout and strain gauge arrangement.

### 2.3. Numerical Simulations

The slope-tunnel deformation characteristics of the upper slope before and after the occurrence of sliding were analyzed using FLAC3D, and the model test results were verified.

The numerical model consists of 44,728 elements and 10,673 nodes. The size of the numerical model was reduced to 1:1 with the size of the test model. The bedrock part of the model was 200 cm long, 100 cm wide, and 45 cm high; the landslide part was 200 cm long, 100 cm wide, and 35 cm high; the tunnel portal retaining wall was 50 cm wide and 15 cm high; the diameter of the tunnel was 20 cm, as shown in Figure 4. The boundary conditions

were fixed at the bottom of the model; the front and rear boundaries and the left and right boundaries of the model took normal constraints; and the upper part was a free boundary. The mechanical parameters were selected according to the mechanical parameters in the model test. The elastic, perfectly plastic Mohr–Coulomb model was used to describe the stress-strain behavior of the soil materials, whereas the linear elasticity model was used for the tunnel and tunnel portal retaining wall. The compression region with the same test conditions was used to simulate the slip of the upper soil by directly applying the initial velocity to the nodes in the region.

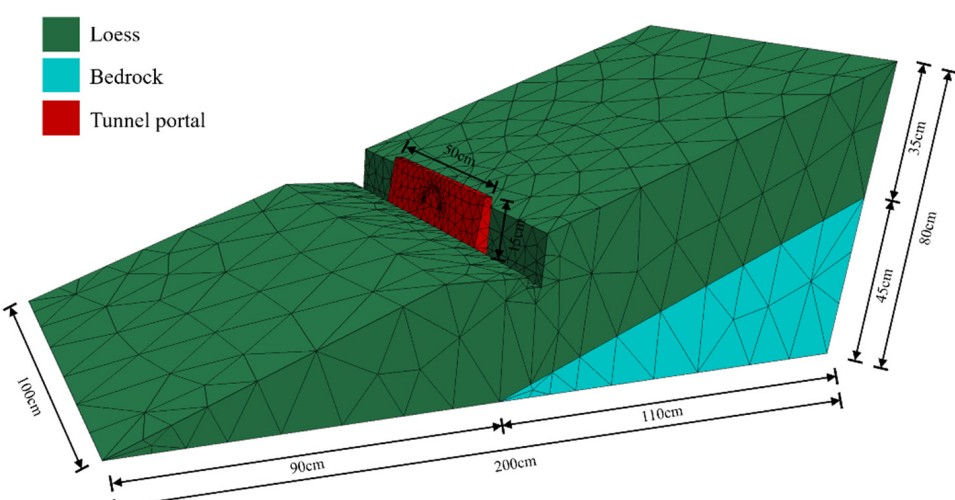

**Figure 4.** Numerical modeling of the test model at a 1:1 ratio.

The main calculation steps of the numerical simulation were as follows: Step 1, initial equilibrium: assign parameters to the slip body, tunnel, and bedrock respectively, and assign gravity load to the model to obtain the initial equilibrium stress field under gravity load. Step 2, simulation of sliding: the initial motion velocity was applied to the model test loading area to simulate the phenomenon of a landslide sliding along the contact surface of the loess bedrock, and a fixed time step was set for the model to solve the operation.

## 3. Results

### 3.1. Mechanical Behaviors and Displacements

In this study, five repeated loading tests were conducted, with each loading test conducted for 20 minutes unloading. Figure 5 shows the load–displacement and displacement–time curves of the MTS. There are five curves in Figure 5a, which represent two different deformation modes: a long "S" shaped curve (o-a), and a short "S" shaped curve (a-b, b-c, c-d, d-e). The change curve has three stages: the compaction stage, the deformation accumulation stage, and the sliding stage. The difference between the two modes lies in the compaction stage. In the curve o-a, compaction is maintained for a long time, the porosity in the soil decreases, and the soil becomes consolidated. After compaction, the interface between loess and bedrock is affected by friction. The soil slides along the interface when the friction reaches the maximum static friction force (60 kN). The first inflection point appears on the displacement–time curve. When unloading, the displacement remains unchanged. In curve a-e, the compaction time is greatly reduced, and the deformation of soil is mainly due to slippage. The maximum force of each curve is close to 45 kN and the maximum displacement is 18 mm. According to the displacement–time curve, the soil shows complete plastic behavior.

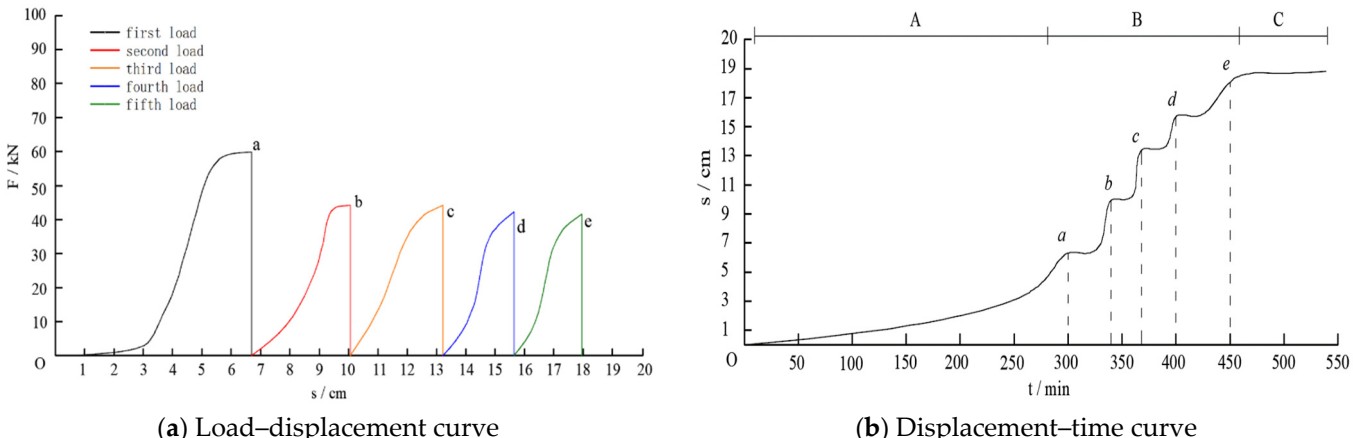

(**a**) Load–displacement curve      (**b**) Displacement–time curve

**Figure 5.** MTS data measured during the loading process.

Therefore, we assume that the loading method was displacement controlled. The residual curve appeared under continuous loading conditions with a value of 60 kN. In this experiment, the peak force decreased after repeated loading. This result indicates that the maximum static friction force was 45 kN. The first peak force contains the cohesive force between the soil particles, as well as the resistance caused by the tunnel. In conclusion, the initial slip had the largest force, and once the slope had slipped, the next slip occurred easily.

The tunnel moved as the slope slipped. The laser cross site at different time points was used to reflect the tunnel deformation "*l*". The original site was regarded as the zero point, and the tunnel photo was measured every 30 min. The tunnel deformation characteristics are shown in Figure 6. In the first 300 min of the test, the image showed a minimum pulsating deformation, and the displacement in the y direction was about 1 cm. After 300 min, the soil deformed, resulting in an obvious deformation of the tunnel. The tunnel displacement increased with the increase in load, and then rebounded after unloading. With the increase in time, the tunnel rebound deformation decreased gradually.

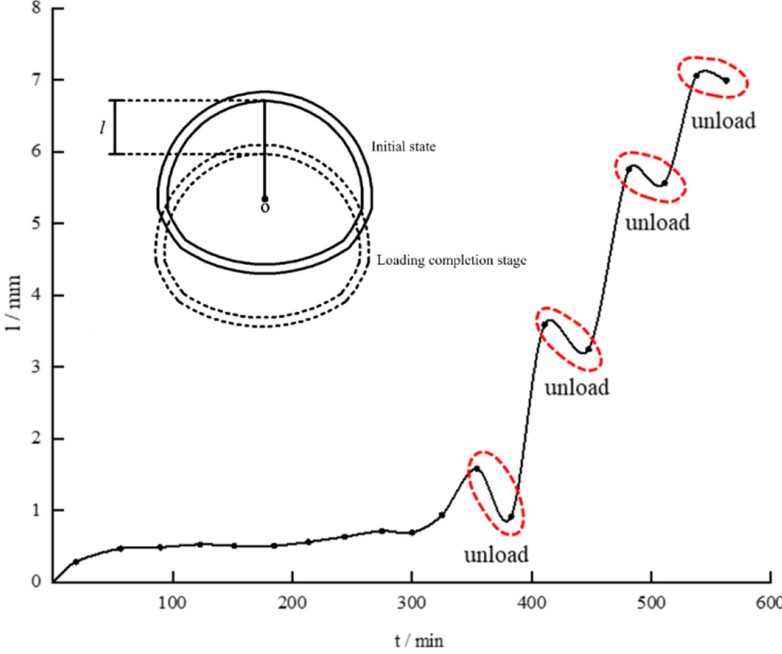

**Figure 6.** Tunnel deformation reflected by the site change of laser cross site.

There are three forces that determine the tunnel's resistance to the landslide: the pressure upon the soil, the support force of the underlying soil, and the embedded force of

the bedrock. The test results show that the supporting force of the lower soil increased with the decrease in the influence of the embedded force. A series of annular cracks developed in the tunnel structure.

In order to compare the soil pressure changes at different slope heights, the earth pressure cells along the axle wire (1-2, 2-2, 2-5, 2-8, 3-1, 3-2) were regarded as the objects. There were three zones, as illustrated in Figure 7: A is the soil upon the tunnel, B is the tunnel, and C is the soil below the tunnel. The soil pressure was zero in the B zone. There were four measurement points at heights of 80, 50, 30, and 10 cm. The soil pressure was 400 kPa at 80 cm, which is equal to the MTS's force. The soil pressure was 268 kPa at 50 cm, 110 kPa at 30 cm, and 60 kPa at the bottom of the slope. The soil pressure of uniformly distributed soil calculated based on the self-weight of soil is shown in the solid line in Figure 7. The theoretical calculated value is the soil pressure of homogeneous soil, while the experimental and numerical calculated values are the soil pressure in the presence of the tunnel structure soil. The soil pressure of uniform soil was lower on the tunnel, but greater below the tunnel. There were significant differences at the 50 cm height. Due to the existence of the tunnel, the transmission of soil pressure was resisted, and the soil pressure was concentrated on the roof of the tunnel. The soil pressure under the tunnel decreased sharply. The curves of the numerical simulation results are in good agreement with the curves of the experimental results. However, the difference in the earth pressure between the uniform soil and the tunnel soil is small. It can be concluded that the main influence on the tunnel was the soil.

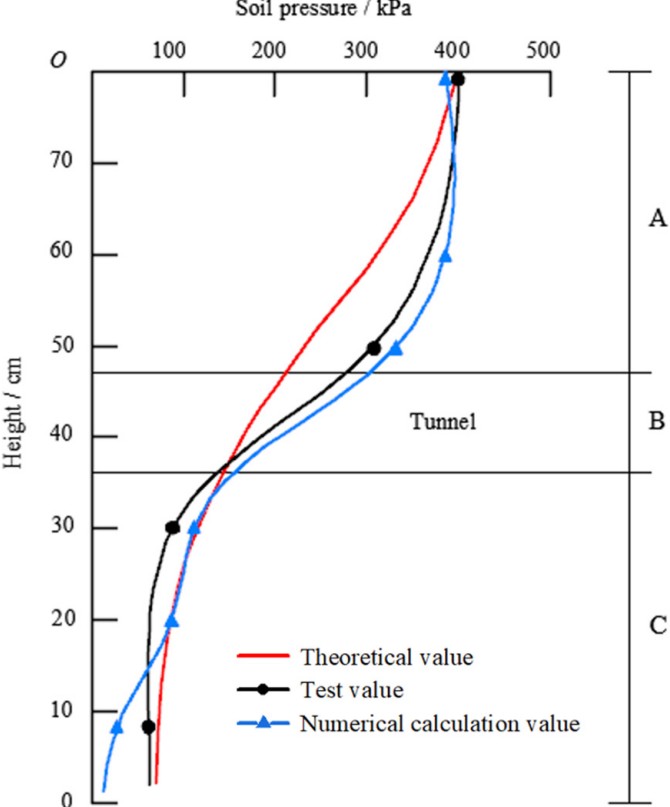

**Figure 7.** Variation Schematic diagram of the variation of soil pressure with slope height.

### 3.2. Lateral Deformation

The lateral deformation of the slope was measured using a PIV system. The horizontal (H) and vertical (V) deformations at different time points are shown in Figure 8.

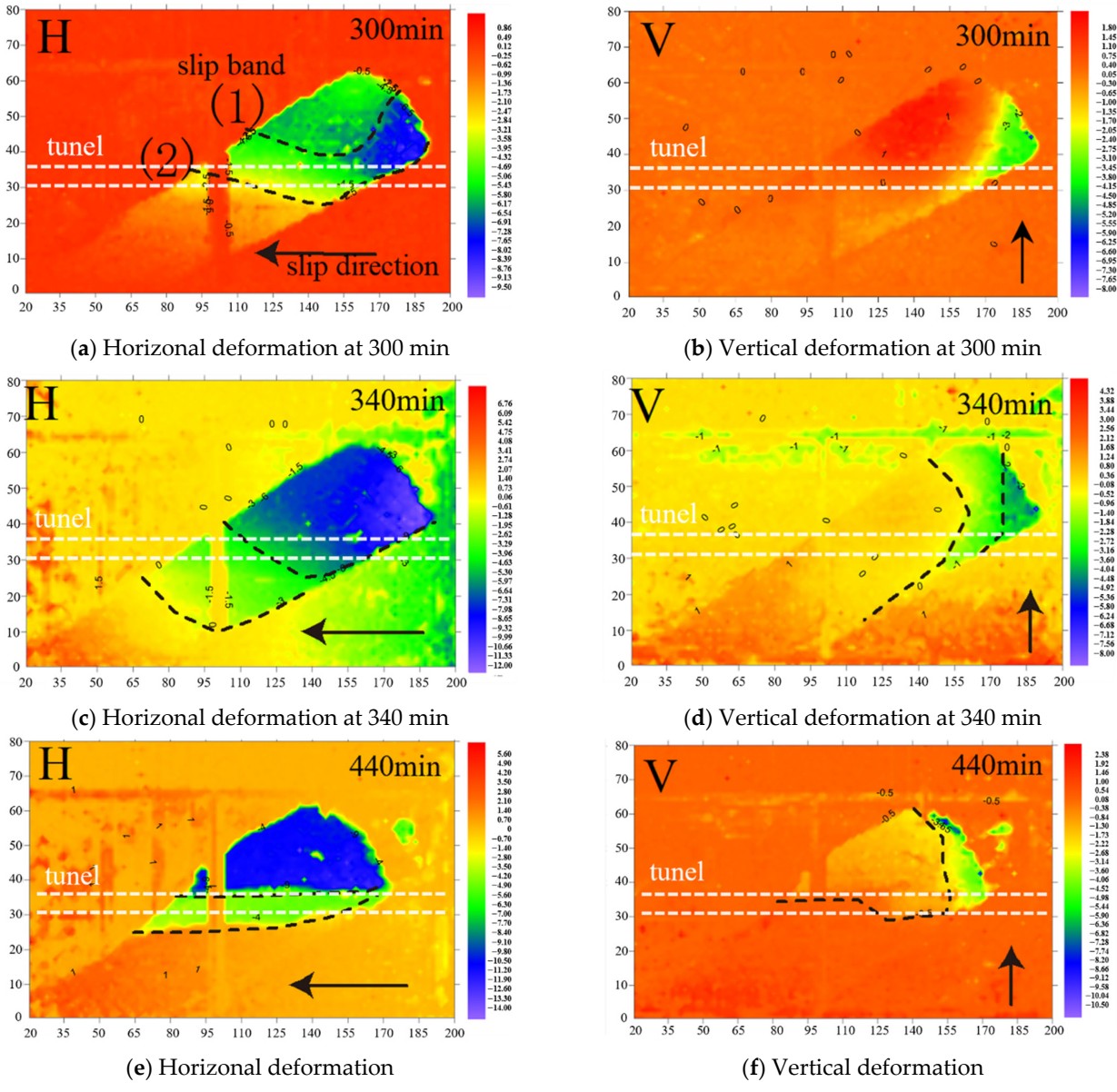

**Figure 8.** Lateral displacement contour measured by PIV system. (Unit: cm).

At 300 min, the soil deformation was obvious. There are three deformation zones on the horizontal contour: the tunnel roof, the tunnel bottom, and the loess-bedrock interface. These deformation zones determine the deformation of the entire slope. The maximum deformation in the horizontal direction was 8 cm, and the deformation in the vertical direction was 3.8 cm, indicating that the soil moved horizontally under the action of trailing edge pressure. The slope above the tunnel presented progressive failure characteristics, and the deformation decreased gradually from the back edge to the front.

At 340 min, different changes can be observed in the three slide bands. The loess-bedrock interface continues to exhibit a linear pattern. The deformation of the tunnel roof and the tunnel bottom bands increases and the band sites change because of tunnel's deformation. The H deformation was 9.32 cm, and the V deformation was 4.48 cm.

At the foot of the slope, the deformation occurs in the direction opposite to the direction of the force load. This phenomenon indicates that the frictional force between the soil and bedrock was greater than the load at the back edge of the slope. The slip zone (1) was completely cut through at 440 min, the deformation zone above and below the tunnel was most clearly defined near the end of the last loading phase, and the deformation of the soil

on both sides of the tunnel exhibited a clear difference. The soil above the tunnel slid as a whole and the bottom soil was compressed. The maximum displacement of the subsoil was 6 cm at the foot of the slope. After repeated loading, the soil above the tunnel showed obvious landslide characteristics: frontal uplift, a slip zone, soil loosening, and collapse. The landslide deformation was greatest in the upper part of the tunnel, with a maximum displacement of 18 cm, while the landslide deformation in the lower part of the tunnel was smaller, with a maximum displacement of 6 cm. The presence of this tunnel reduced the soil deformation threefold.

The horizontal displacement field calculated by the numerical simulation exhibits good agreement with the images taken by the test PIV. The horizontal displacement of the side slope is divided into two parts by the tunnel structure at the top and bottom. Comparing Figure 9a with Figure 9b, it can be observed that the existence of the tunnel structure resulted in significant differences across the slope displacement area. This difference was especially marked at the rear edge of the slope above the tunnel, which is the position where the initial velocity of the model was applied, and also the position where the displacement was the most obvious.

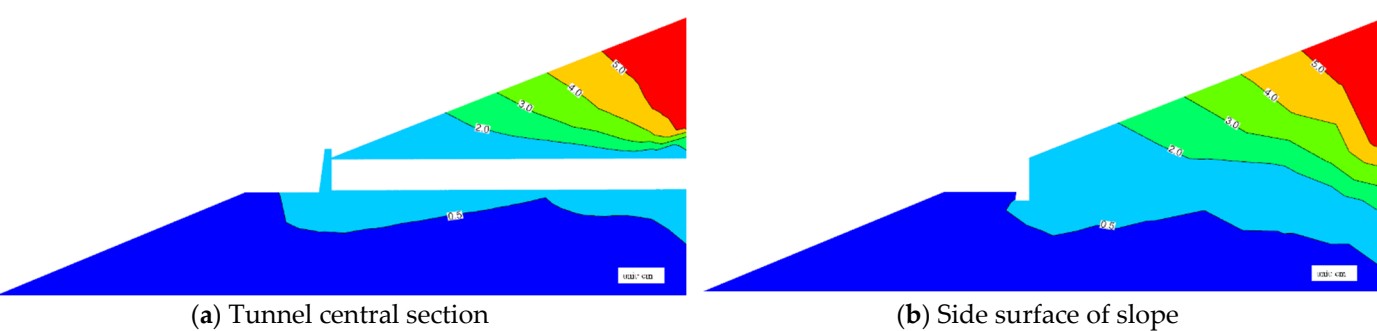

(**a**) Tunnel central section　　　　　　　　　　　　　　(**b**) Side surface of slope

**Figure 9.** Soil displacement field after loading.

### 3.3. Deformation of Loess-Bedrock Interface

The deformation behavior of the loess-bedrock contact zone was monitored by strain gauges in real time, and the landslide was analyzed layer by layer from the back edge to the front edge. In the deformation–time curve, there is an obvious step-like change, which indicates that, in the process of landslide deformation, the energy on the loess-bedrock contact surface accumulates and releases, resulting in an instantaneous deformation.

The deformation increment and duration at different stages can be used to characterize the energy change at the loess-bedrock interface. The larger the deformation increment, the longer the deformation duration, indicating that more energy accumulated during the deformation process. The total deformation increment represents the total energy accumulated by the landslide, which can be used to judge the sliding trend of the landslide. In addition, the different sliding time of the soil indicates the deformation speed and deformation stage of the landslide.

When the strain is negative, the strain gauge is compressed, and the soil movement direction is consistent with the thrust direction. While the strain is positive, the strain gauge is strained, and the deformation is in the opposite direction to the thrust (Figure 10). In the 7th layer, the deformation increased negatively due to the pressure from the trailing edge in the early stage. As the sliding body moved downward and reached the tunnel roof, it was constrained and the deformation decreased, which caused soil accumulation. In addition, there was an upward reaction force on the overlying soil, which caused the deformation of the 7th layer to increase positively. The 6th layer is located at the tunnel, where the soil was in a state of compression, and the deformation increased positively. Layers 1–5 are located at the bottom of the tunnel, and here the soil continued to move downward, and the deformation continued to show a negative growth trend. Over time, the load changes show fluctuation characteristics, indicating that the slope continued to slide, and that the trailing edge of the slope repeatedly separated from and contacted with the loading system.

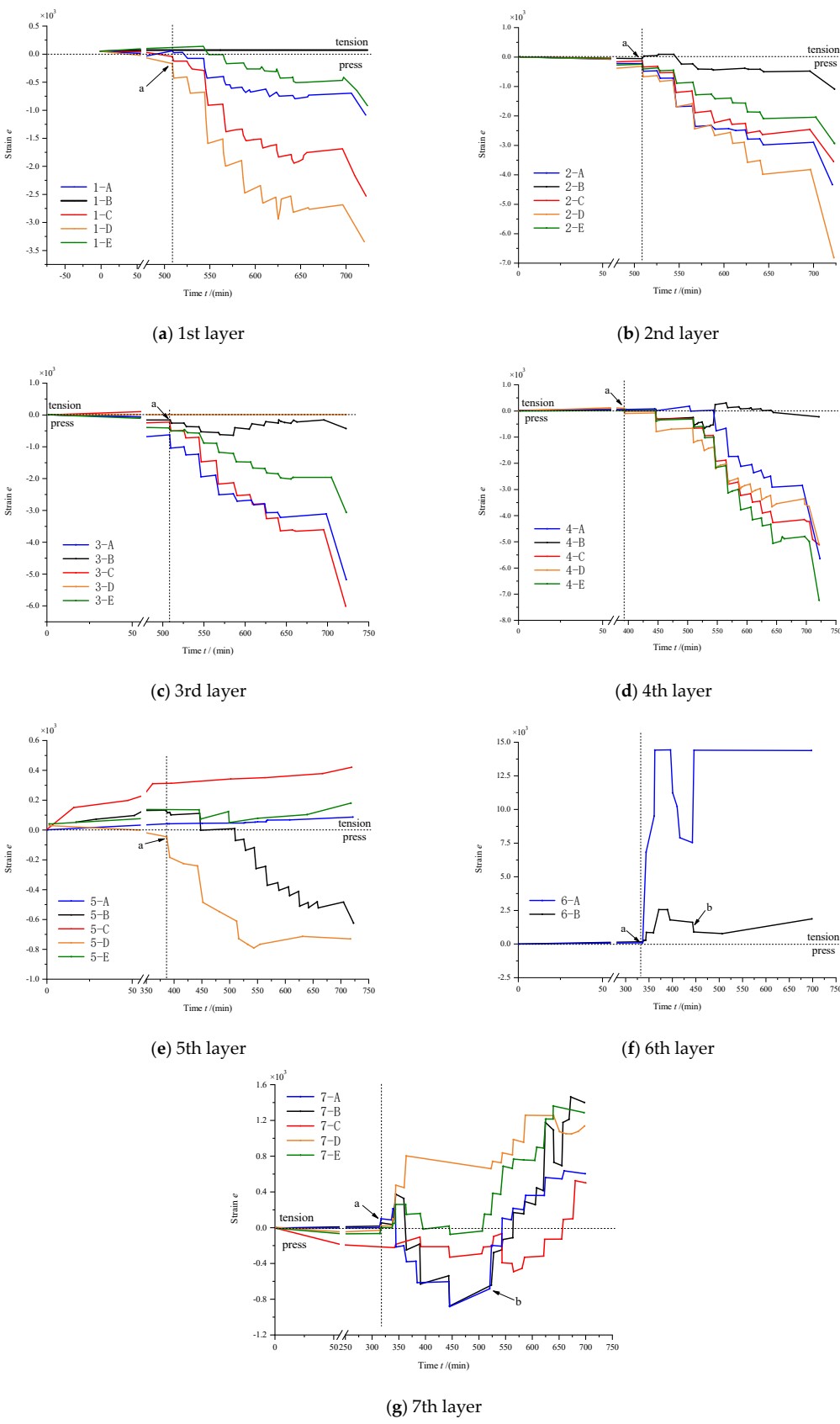

(**a**) 1st layer

(**b**) 2nd layer

(**c**) 3rd layer

(**d**) 4th layer

(**e**) 5th layer

(**f**) 6th layer

(**g**) 7th layer

**Figure 10.** Strain monitored by strain gauges in the loess-bedrock contact zone. The distribution of the location of each layer is shown in Figure 3a.

The deformation data regarding the loess-bedrock interface at different times were collected, and the deformation contour of the interface is shown in Figure 11. At 30 min, the trailing edge was thrust and deformed downward, and the deformation in the tunnel and surrounding soil was zero. With the increasing movement of the trailing edge, the soil near the tunnel began to deform. The deformation of tunnel sides was large. At 270 min, a small deformation began to appear at the leading edge of the soil. With the passage of time, the amount of deformation gradually increased and the range of deformation gradually expanded. At 360 min, the upper soil was completely deformed. When it reached the top of the tunnel, the soil was subjected to the tunnel reaction force and tended to move upward, so the direction of movement gradually developed to the back edge of the slope. At 600 min, the reaction force of the tunnel on the soil was higher than that of the thrust, and the soil began to exhibit an extrusion deformation at the trailing edge. The location of the tunnel became the boundary of the displacement deformation. The upper soil of the tunnel moved upward, mainly manifested as soil uplift. In the lower part of the tunnel, the soil continued to move downward, showing a sliding failure mode.

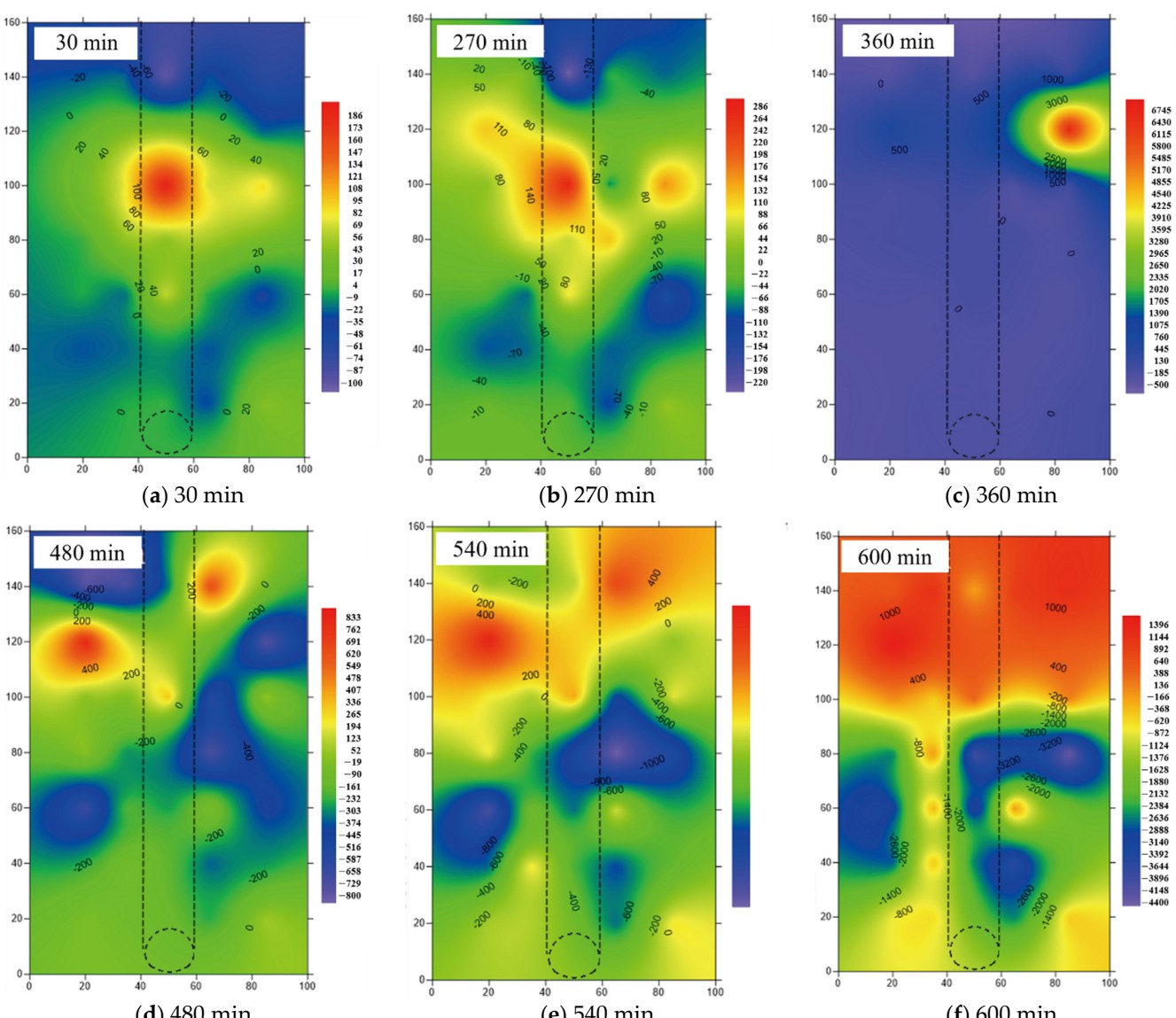

**Figure 11.** Loess-bedrock interface deformation contour measured by strain gauges 600 min into the test.

The simulated displacement contour changes in the loess-bedrock interface are shown in Figure 12. When the velocity was applied to the back edge of loess, the soil near the

tunnel began to deform, but the deformation was mainly on both sides. The landslide front under the tunnel was also deformed, and the deformation range was large, especially on both sides of the tunnel front; this occurred at the same time as in the test results, and the displacement increased significantly. The reason for this is that the tunnel structure, as the boundary, played a supporting role for the soil above, while it had less influence on the soil below, resulting in a more obvious downward sliding deformation of the soil below. Comparing these results with those illustrated in Figure 11f, measured by the buried strain gauges during the test, the deformation law can be seen to be roughly consistent.

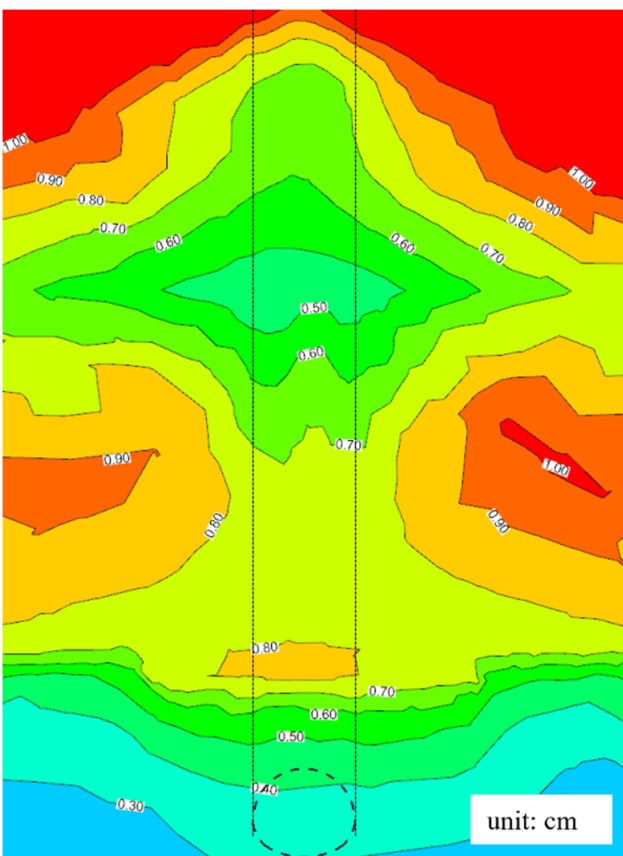

**Figure 12.** Loess-bedrock interface deformation contour calculated by the numerical simulation.

The numerical simulation results show that the soil on both sides of the cave entrance was deformed in the middle (Figure 13a), indicating that although the soil should have slid downwards along the loess-bedrock sliding surface during the test, the presence of the tunnel structure meant that the direction of the loess movement was changed, and that the soil above the tunnel rose while the soil on both sides was deformed in the middle. Figure 13b also shows that the vertical deformation of the soil on both sides of the tunnel was clearly larger than that of the soil above the tunnel structure.

The tunnel changed the movement path of the soil, making the soil on both sides move to the middle, and the soil exhibited a linear uplift and parallel cracking, as shown in Figure 14a. Through lateral observation and comparison with the initial state (Figure 14b), the uplift deformation of the upper soil of the tunnel can be clearly seen, whereas the bottom uplift was small.

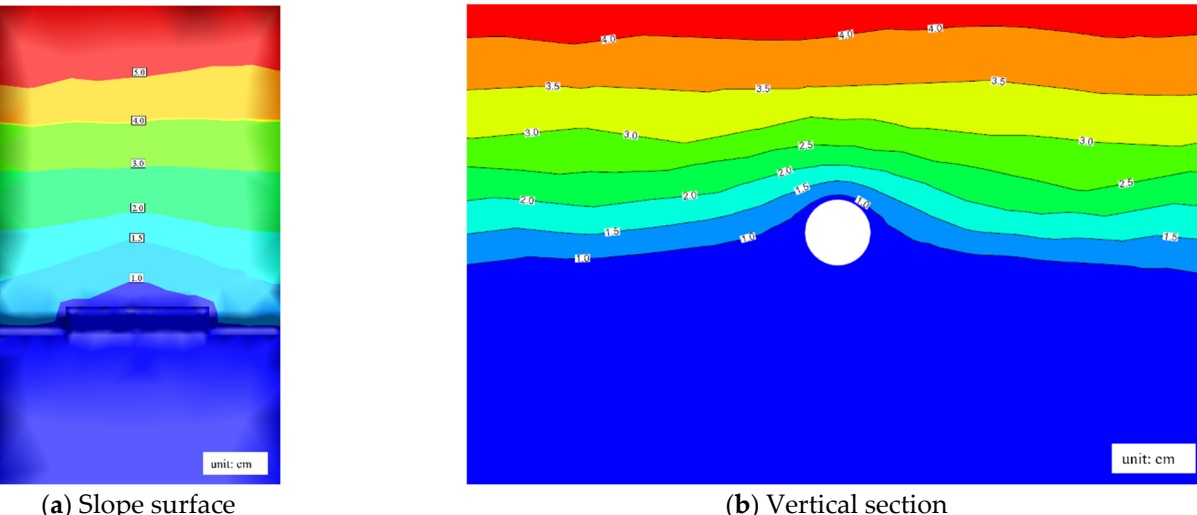

(**a**) Slope surface  (**b**) Vertical section

**Figure 13.** Deformation contour of slope surface and vertical section calculated by the numerical simulation.

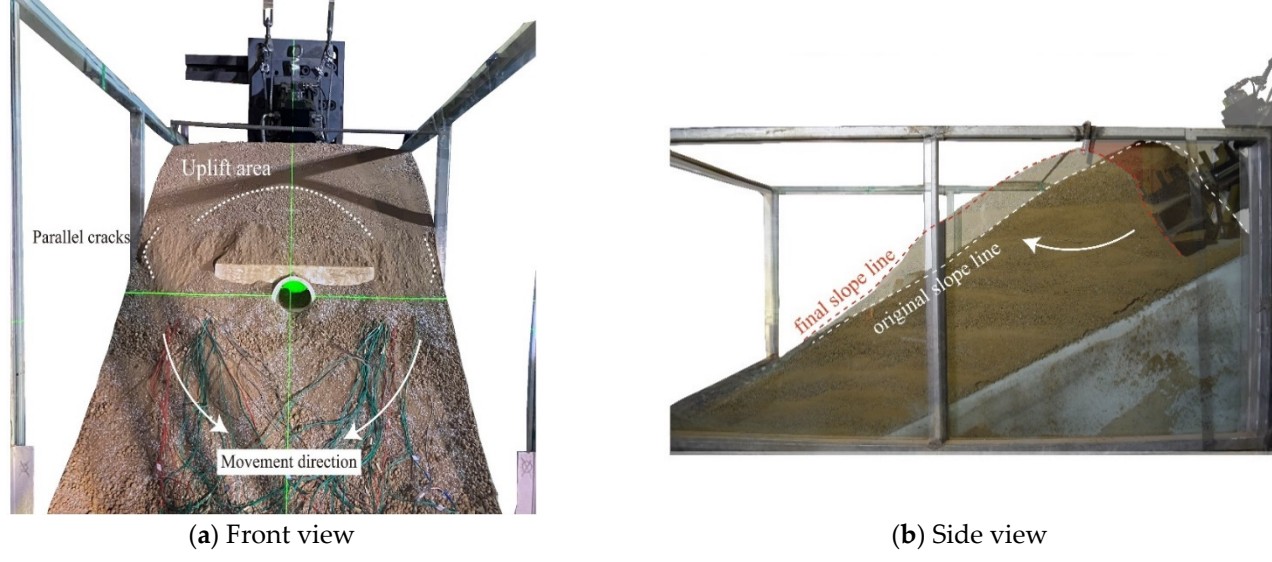

(**a**) Front view  (**b**) Side view

**Figure 14.** Deformation characteristics of the slope at the end of the test loading.

## 4. Discussion

The sliding zone of a loess-bedrock landslide is located on the contact surface between the loess and bedrock. At the contact surface of relatively flat sedimentary rock, the main deformation mode is friction sliding, and the friction law dictates the deformation process. This movement characteristic determines that this type of landslide will have periodic sliding, namely stick-slip, which is a process of energy accumulation and release. When the deformation is stable, the soil compacts gradually from the back to the front. When the compactness reaches a certain degree, the accumulated potential energy in the soil reaches a critical value, and the potential energy is transformed into kinetic energy, so that the soil slides. This process is short and rapid. When a certain amount of potential energy is released, the potential energy of the soil and the potential energy generated by friction are balanced, and the soil reaches a static state again and begins to accumulate energy.

Because soil is a porous elastoplastic media, compaction, elastic deformation, plastic deformation, and whole slip occur during the process of deformation, in which both elastic deformation and plastic deformation occur simultaneously. The soil stress decreases gradually with the increase in depth, and the tunnel affects the stress transfer and changes the stress state of the soil, and further influences the movement mode of the landslide. The

essence of the shear stress in soil lies in the friction force and bite force between soil particles. After multiple cyclic shearing, the soil particles in the sliding zone, which were originally arranged in order during the consolidation stage, re-interlock with each other. Hence the shear strength of the first shearing is the greatest. As the particles are gradually arranged in order during the repeated shear process, the shear stress displayed by the particles gradually decreases, and the shear stress tends to a constant value after the completion of the particle arrangement, which is the shear residual stage.

The spatial relationship between a landslide and a tunnel will show different deformation and failure forms. According to the landslide characteristics and tunnel distribution, there are seven different types (Figure 15). Among them, c and d are tunnels through bedrock, and their stability is not affected by landslides. Type a and f cross the slide body in different directions, and the tunnel structure is completely located within the slide body. During slope sliding, whole deformation of these two kinds of tunnels will occur along with the soil mass. The deformation characteristics of these two kinds of tunnels can be determined by using the beam model of elastic foundation [26–32]. In the loess area, the deformation of the shallow surface is the main form of landslide, playing a large role in slope collapse [33,34]. A shallow surface landslide changes the stress characteristics of the tunnel. For an f-type tunnel, on the one hand, it will produce a bias voltage, and on the other hand, it produce a friction effect between the soil and the tunnel structure. For a g-type tunnel, due to the change in the earth pressure behind the wall, wall cracking often occurs, and sometimes the tunnel is buried by a mud flow. The e-type tunnel is an extreme case, in which the sliding zone passes through the tunnel structure under shear action, leaving it prone to deformation and failure. Faults or structural planes pass through tunnels in similar ways within the rock mass. This kind of tunnel does not appear in engineering in general, and is analyzed and discussed as a theoretical model.

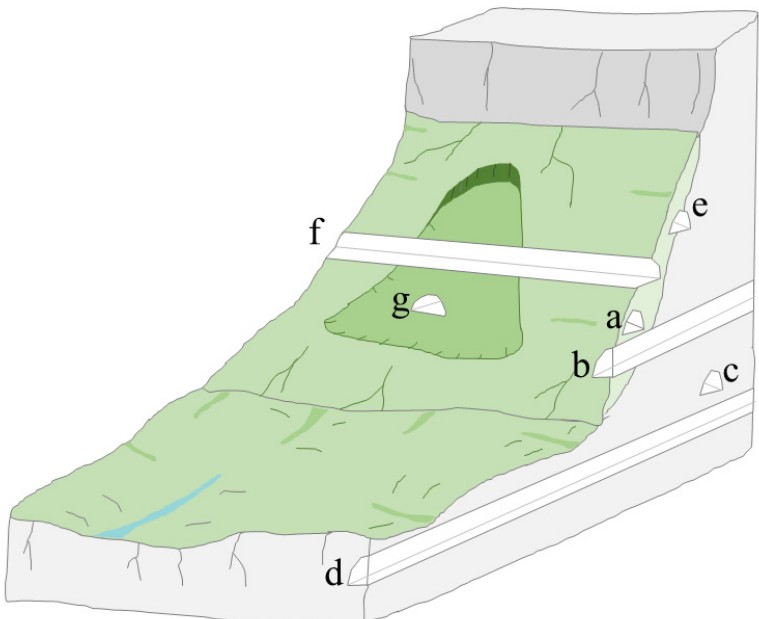

**Figure 15.** Tunnel–landslide spatial relationship. (a~g: The spatial relationship between a landslide and a tunnel.)

In this paper, type b was taken as the study object. In this case, the tunnel is partly embedded in the bedrock and partly in the loess. According to the force analysis, the tunnel is mainly subjected to three kinds of forces: pressure of the upper soil on the tunnel structure G, the supporting force of the soil at the bottom N, and the embedded force of the bedrock on tunnel F. The pressure G is jointly determined by the self-weight of soil and the friction between the loess and bedrock. When water infiltrates the soil, its self-weight is superseded by its effective weight. Loess-bedrock friction is related to the

soil properties and interface morphology [35–37]. The supporting force N of the soil at the bottom is similar to the bearing capacity of the foundation and is related to the compression coefficient and stress history of the soil. In addition, the friction between the underlying soil and the bedrock also provides a part of the supporting force. The lower soil can be simplified as a mechanical model of an elastoplastic foundation beam by considering the elastoplastic characteristics of the soil, which need further research in the future. Because the tunnel is partially embedded in bedrock, it forms a cantilever beam structure, which is subjected to shear and bending deformations. This force is related to the structural characteristics of the tunnel, including its rotational inertia, elastic modulus, and other parameters. These three forces work together to control the deformation of the tunnel.

## 5. Conclusions

Based on the laboratory model test combined with numerical simulations, the deformation characteristics of a loess-bedrock landslide through a tunnel were studied, and the lateral and contact surface deformation of the sliding landslide were monitored. The main conclusions are as follows:

(1) The anti-sliding force of the loess-bedrock landslide around the tunnel comes from the maximum friction force of the contact surface, the cohesion force of the soil particles, and the resistance of the tunnel. The force required to cause a landslide is relatively large, and the landslide slip resistance decreases after the initial slip, so that subsequent soil displacement slips are more prone to occur during the landslide.

(2) The tunnel changes the distribution characteristics of the soil pressure within the landslide, whereby the soil pressure at the top of the tunnel increases, which has a negative impact on the tunnel structure. There are three types of landslides based on the three different slip zones: the loess-bedrock landslide, the slide above the tunnel, and the shallow slide at the sides of tunnel.

(3) The deformation process of a loess-bedrock landslide is sudden, and is a process of energy accumulation and release. After the sliding of the existing sliding body below the tunnel, parallel cracks will be formed on both sides of the tunnel, and the soil will tend to gather in the middle.

**Author Contributions:** Conceptualization, Z.S. and X.Y.; methodology, S.L.; software, X.Y.; validation, Z.S., Y.C. and P.L.; investigation, Y.C.; data curation, S.L.; writing—original draft preparation, Z.S.; writing—review and editing, P.L.; visualization, Y.C.; supervision, P.L. All authors have read and agreed to the published version of the manuscript.

**Funding:** This research was funded by the National Key R&D Program of China, grant number 2018YFC0809600, the National Natural Science Foundation of China, grant number 51978019 and the Beijing Natural Science Foundation, grant number 8222004.

**Institutional Review Board Statement:** Not applicable.

**Informed Consent Statement:** Not applicable.

**Data Availability Statement:** The relevant data are all included in the paper.

**Conflicts of Interest:** The authors declare no conflict of interest.

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
