# Peer review of "Influence of a Landslide on a Tunnel in Loess-Bedrock Ground"

_applsci, doi:10.3390/app12136750_

Round 1

Reviewer 1 Report

The topic sounds interesting, The experimental works seems good, but the structure of the paper must be improved.

1. Abstract and introduction should be rewritten to indicate the problem statement, originality, objective.

2. Abstract should include clear result and conclusions

3. Please provide properties used in the laboratory testing

4. Please check figure numbering. Some figures have wrong figure number

5. Figures are too much. Please shorten it into 15 figures only

Author Response

Please find the authors' reply in the attachment.

Reviewer 2 Report

The presented work is well organized and novel. The manuscript may be checked from English Grammar point of view

Author Response

(The authors gave the same response as above.)

Reviewer 3 Report

*Overall summary: 

The manuscript considers laboratory tests and numerical simulation of landslides on tunnel behaviour in loess and bedrock.  The technical content of the paper is sound and suitable for publication.  The language requires significant improvement prior to publication.

* Comments and observations:

Detailed review and elaboration of are presented below, with distinction between general (ge), technical (te), and editorial (ed) comments regarding the manuscript.

(ed) A number of sentences are poorly written and require modification.  Examples include lines 34-35, lines 43-44,

(ge) Lines 45 – 46: what type of landslide are you talking about here?  Many landslides do not occur after vibration.  Please be specific.

(te) Lines 54 – 55: What type of numerical simulation?  This seems a rather vague claim.

(ed) Line 69: Researchers pay more attention on the results but the progress.  This sentence adds very little in the way of detail to the paper.  Suggest removing.

(te) Section 2.3 provides a somewhat limited description of the numerical model.  Further details are suggested.

Author Response

(The authors gave the same response as above.)

Reviewer 4 Report

The content of the article is very important for tunnel excavation in the bed ground. The article was written correctly.  I have no reservations about the merit value. I have found some minor remarks which are listed below. 

Avoid leaving single letters at the end of the line, e.g. line 54, and single words at the end of a paragraph, e.g. page 2, paragraph 1, and page 6, paragraph 2.

There is no reference in the text to formula (1).

The values should be together with the units in one line, e.g. line 106 - 20 cm and line 191 - 50 cm.

Figures should be included in the text after they are cited. Lack of reference and description in the text for Figures no. 10, 12, 13.

Drawing no 15 and 16 should be attached entirely on one page, not split.

There should be a dot on line 305 after Fig. 17 and on line 318 after "middle" there should be a space.

Correcting these shortcomings should improve the quality of the article and its better reception.

Author Response

(The authors gave the same response as above.)

Round 2

Reviewer 1 Report

The manuscript has been revised accordingly. The manuscript can be accepted as is